# Staged Surgery for Intra-Extracranial Communicating Jugular Foramen Paraganglioma: A Case Report and Systematic Review

**DOI:** 10.3390/brainsci12091257

**Published:** 2022-09-16

**Authors:** Qiang Li, Yanbing Yu, Li Zhang, Jiang Liu, Hongxiang Ren, Xueke Zhen

**Affiliations:** 1Department of Neurosurgery, China-Japan Friendship Hospital, Beijing 100029, China; 2Department of Neurosurgery, Baotou Steel Group Third Staff Hospital, Baotou 014010, China; 3Graduate School, Chinese Academy of Medical Sciences & Peking Union Medical College, Beijing 100006, China; 4Graduate School of Peking University Health Science Center, Beijing 100191, China; 5Graduate School of Capital Medical University, Beijing 100069, China

**Keywords:** intra-extracranial communicating jugular foramen paraganglioma (IECJFP), staged surgery, LCNs deficit and tympanic cavity involvement, radiotherapy, “wait and scan” strategy

## Abstract

Staged surgery strategy was preferred for patients with intra-extracranial communicating jugular foramen paraganglioma (IECJFP). A female patient who presented mild tinnitus, headache, and dizziness, together with preoperative related imaging, was diagnosed with a left intra-extracranial communicating jugular foramen lesion in November 2015 and accepted an initial operation for the intracranial tumor by retrosigmoid approach. The pathologic report was paraganglioma. In November 2021, a subtotal resection of the extracranial tumor was conducted for prominent lower cranial nerves (LCNs) deficit and middle ear involvement by infratemporal approach. In patients with IECJFP accompanied by LCNs deficit and middle ear involvement, an initial surgery for extracranial lesion and a second procedure for intracranial tumor were appropriate. However, the first operation for the intracranial lesion was preferred in IECJFP cases without LCNs deficit and middle ear involvement, as it could remove compression to the neurovascular structure and brain stem, clarify a pathological diagnosis, avoid a CSF leak, and prevent a severe neurological disorder from extracranial lesion excision. Subtotal resection of the extracranial tumor would be performed when lesion became larger combined with obvious LCNs disorder and tympanic cavity involvement. Consideration of specific staged surgical strategy for IECJFP in accordance with preoperative LCNs deficit and tympanic cavity involvement could prevent critical postoperative neurological deficit and improve quality of life in the long term.

## 1. Introduction

Paraganglioma is a kind of tumor rich in blood vessels, which can arise in different parts of the paraganglion system. Approximately 3% of extra-adrenal paragangliomas (PGs) were found in the head and neck, where PGs originated frequently from carotid bodies, jugulotympanic paraganglion cells and vagal nerves with low incidence at 1:30,000 in head and neck tumors [1,2]. Although the PGs were usually benign and grew slowly [3], presentations in patients with jugular foramen paragangliomas (JFPs) were common and complex [4]. Almost one third of JFPs produced intracranial involvement [5]. Surgery for JFPs faces enormous challenges, but it was widely recognized that JFPs ought to be removed by distinct surgical strategies [5,6,7]. Referring to intra-extracranial communicating jugular foramen paragangliomas (IECJFPs), staged surgery would be the proper therapeutic modality, especially with larger intracranial tumors [8].

## 2. Clinical Case

A 48-year-old female patient consulted a neurosurgeon due to mild tinnitus, headache, and dizziness in November 2015. A significantly enhanced intra-extracranial communicating jugular foramen lesion was found in the left enlarged jugular foramen on contrast-enhanced magnetic resonance images (CE-MRI), and attached to the brain stem and cerebellum (Figure 1(1),(2)). The intracranial lesion was totally removed through the left retrosigmoid approach, then preoperative presentations were relieved without new postoperative complications. Postoperative CE-MRI showed total resection of the intracranial tumor (Figure 1(3),(4)), while the extracranial lesion (Figure 1(5),(6)) employed the “wait and scan” strategy. The pathologic result was paraganglioma [2] (Figure 2(1)) without malignancy, in which chromogranin A (CgA), S100, and Synaptophysin (Syn) were positive immunohistochemically (Figure 2(2),(4)).

In October 2021, the patient found that the extracranial lesion had enlarged, accompanied by left hearing loss, hoarseness, dysfunctional deglutition, protruding tongue towards left with ipsilateral atrophy, left shoulder muscles paroxysmal contraction, and resistant hypertension. Magnetic resonance images and CE-MRI showed that the extracranial lesion with vascular flow void (Figure 3(1)) in the left jugular foramen became enlarged and highly enhanced (Figure 3(2),(3)). Computed tomography angiography presented that feeding vessels of the extracranial lesion arose from branches of the external carotid artery (Figure 3(4),(5)), and drainage veins returned to the external jugular vein (Figure 3(6)). The extracranial lesion was closely associated with the internal jugular vein which was invaded and occluded by the tumor (Figure 3(7),(8)). A second surgery was performed for the extracranial lesion through the infratemporal approach. Cranial nerves and vessels around the lesion were preserved completely. Postoperative blood pressure remained normal without antihypertensive drugs, and critical preoperative lower cranial nerves (LCNs) deficits (dysphagia, dysphonia, abnormal tongue movement, and stimulated shoulder muscles contraction) were relieved. The postoperative CE-MRI displayed that most of the extracranial tumor had been resected, but less residual lesion was detected in the jugular foramen (Figure 4(1),(3)). Aside from traditional pathologic presentation of PGs in the second surgery, the supplementary SDHB and GATA3 were characteristically positive.

The patient accepted radiation for the residual tumor when discharged over two weeks. In the three follow-up months, more improvements in LCNs deficits were observed, and blood pressure was stable without drugs. The patient had gone into work and attended social activities with a better quality of life.

## 3. Discussion

Jugular foramen paragangliomas (JFPs) are commonly benign and steady tumors, but JFPs are also involved in important surrounding neurovascular structures [2]. Erosion and damage of surrounding bony structure, including jugular plate/foramen and tympanolabyrinthine, in jugular foramen was frequently detected on computed tomography (CT). However, an MRI was the main approach for assessing tumor extension and involvement of adjacent neurovascular structure, where a “salt-and-pepper” signal was the particular manifestation on hypointense T1 or isointense/hyperintense T2 weighted images [9].

According to the Fisch classification for tympanic and tympanojugular paragangliomas [10], class D presents the involvement of posterior fossa intradural extension, but a new proposed class D of Fisch classification removed De component (epidural component). Therefore, a class D paraganglioma also determinately means an intra-extracranial communicating jugular foramen paraganglioma (IECJFP). From a study of complicated tympanojugular PGs [8] where proportion of IECJFP was 75%, IECJFP was considered the most complex lesion in tympanojugular PGs, which would produce more difficulty in surgical operation. Nevertheless, Mazzoni et al. [11] still considered that surgery together with non-surgical therapeutic methods would be the most effective treatment plan.

Surgery for IECJFP has been performed for several years, but incidence of serious postoperative complications was high [5]. In spite of more risks, surgery was still the main treatment approach in order to resect the tumor completely [8,12,13,14], with either a single or staged surgery. Prasad et al. [15] assumed that eradicating glomus tumors completely with reduced LCNs deficits only relied on surgical excision, but they did not present a specific surgical strategy. In this case report, we performed the initial surgery for the intracranial lesion instead of the extracranial tumor, as the patient had fewer and more mild symptoms without LCNs deficit and middle ear involvement. Supposing the first procedure were for an extracranial tumor, the risk of cranial nerves deficit and other severe postoperative complications could be high [8,16], which would seriously worsen quality of life. Considering that important neurovascular structures would be destroyed and the lesion would probably become malignant after long-term radiosurgery for JFPs [17,18], we conducted a “wait and scan’’ policy for the extracranial tumor after the original operation. From a study of “wait and scan’’ strategy in class C and D tumors [15], tumor stable rate was 92% at 3 years and 83% at 5 years. A “wait and scan’’ policy was possibly considered as one of the treatment strategies for IECJFP, especially in patients with functional LCNs before radiosurgical intervention [15,19]. Over six years of “wait and scan’’, patient was satisfied with quality of life until the extracranial tumor enlarged with significant LCNs deficit and tympanic cavity involvement in October 2021.

Jackson et al. [5] found that subtotal resection for PGs involved in the skull base would be the proper therapeutic approach for preservation of LCNs and elderly cases. Sheehan et al. [18] considered that purpose of surgery for glomus tumors was lesion resection as much as possible, rather than total resection. Subtotal excision with preservation of most of LCNs would improve postoperative quality of life in JFPs patients [20]. In this staged surgery, total resection was performed for the intracranial lesion in first operation, and we employed subtotal excision for the extracranial tumor in the second surgery in order to prevent serious postoperative complications. With regard to Fisch class C or D tumors, neither single nor staged surgery resected tumors completely, even though the operation was performed by an experienced neurosurgeon. Furthermore, LCNs preservation made gross total resection more difficult [21]. Yildiz et al. [22] viewed that total resection was more appropriate for class A and B temporal bone paragangliomas, whereas subtotal resection was more applicable in class C and D tumors.

Postoperative LCNs paralysis of JFPs was more frequent than other PGs in the head and neck, whereas the rate of LCNs deficit was nearly 25%–50% [12]. If preoperative cranial nerve paralysis was detected in patients with IECJFP, the intraoperative tumor and cranial nerve were altogether resected in almost 87% of cases [5]. Incidence of cranial nerve palsy was positively correlated with proportion of IECJFP in PGs [5]. From the research about complex JFPs where most were IECJFPs [8], LCNs deficits were detected frequently, in 74.1%, of which cranial nerves IX and X were seen most commonly. For avoidance of a postoperative LCNs disorder, we adopted an initial surgery for the intracranial lesion to remove compression to the surrounding structure and define the pathological result. Furthermore, preoperative headache, dizziness, and mild tinnitus were evidently relieved. When the extracranial tumor was enlarged with noteworthy LCNs disorder and tympanic cavity involvement, we performed subtotal resection for functional reservation of cranial nerves.

Wanna et al. [23] encouraged neurosurgeons to resect the tumor sub-totally in larger JFPs cases in order to avoid serious postoperative neurological dysfunction. Ivan et al. [24] found that postoperative cranial nerve dysfunction perceived in a small number of JFPs patients would bring severe presentation. Preoperative identification from involvement of the skull base, brain stem, cranial nerves, and vessels in radiological examinations could possibly predict and avoid postoperative neurological dysfunction [12]. Ideally, we preserved the structure and function of cranial nerves and avoided damage to important surrounding vessels in the second operation by adequate preoperative radiological assessments. Due to normal blood pressure and improvement of preoperative LCNs deficits, the patient achieved a satisfying quality of life. Although preoperative embolisms for head and neck paragangliomas had been widely recognized [20,25], embolization for IECJFP perhaps raised risks of cranial nerve deficits [26]. We did not employ preoperative embolization in this staged surgery, but intraoperative feeding arteries and drainage vessels were primarily disconnected in the second surgery.

Radiotherapy for JFPs was effective and feasible [27]. Referring to the variation of succinate dehydrogenase (SDH) mutation and the malignant tendency of paragangliomas in the head and neck, radiosurgery was recommended necessarily through related pathologic findings and other particular imaging. Ota, Y. et al. detected that the apparent diffusion coefficient (ADC) and the diffusion-weighted imaging (DWI) were important and useful to differentiate SDH mutation [28,29]. Ktrans as a diagnostic parameter of dynamic contrast-enhanced MRI (DCE-MRI) for PGs [30] could predict the potential efficacy of radiotherapy for paragangliomas [31]. Although radiotherapy for JFPs is widely described in the literature, efficacy ought to be verified for a longer period of time [18,32,33]. Compared with subtotal resection for class C and D tumors, the tumor control rate was higher in subtotal resection plus radiosurgery, at 98% [34]. No evidence supported the notion that radiotherapy would replace surgical resection as an initial therapeutic approach in PGs of classes C and D [15]; additionally, radiological treatment was not able to relieve preoperative cranial nerve deficit radically [27]. As a result, we carried out radiation for the residual tumor after the second surgery in order to obtain long-term tumor control. Our modified staged surgery strategy in this IECJFP patient through assessment about preoperative LCNs deficit and tympanic cavity involvement still demands more samples as well as more time and comparisons to observe and verify.

## 4. Conclusions

We could consider personalized staged surgical strategy together with noninvasive treatment to improve long-term quality of life in IECJFP patients with LCNs deficits and involvement of the tympanic cavity. Initial surgery for the extracranial tumor and a second procedure for the intracranial tumor are appropriate in IECJFP patients accompanied by LCNs deficit and middle ear involvement. However, for IECJFP cases without LCNs deficit and tympanic cavity involvement, the first operation for the intracranial lesion together with the second procedure with subtotal resection for the extracranial tumor would be preferred.

## Figures and Tables

**Figure 1 brainsci-12-01257-f001:**
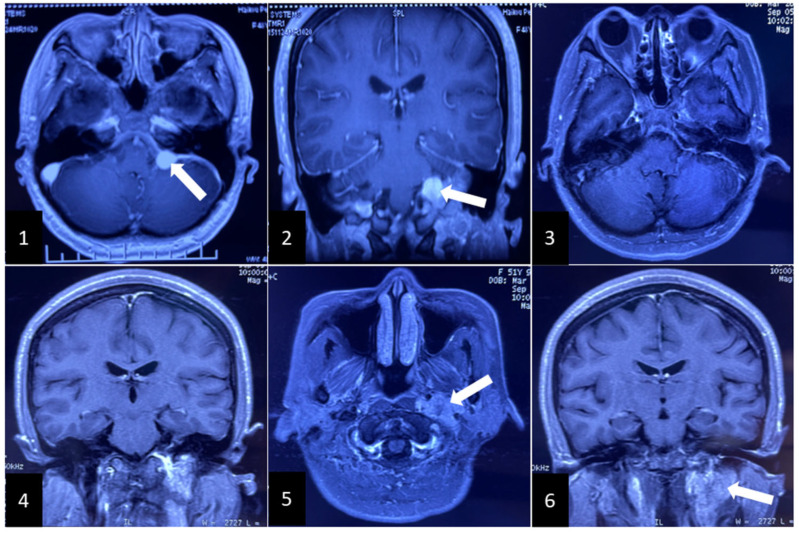
MRI and contrast-enhanced magnetic resonance images (CE-MRI) in the initial surgery. (**1**) and (**2**): displaying a highly enhanced intracranial lesion. White arrow: intracranial lesion; (**3**) and (**4**): presenting total resection of intracranial tumor; (**5**) and (**6**): revealing the residual extracranial lesion in the left jugular fossa and neck. White arrow: residual extracranial lesion.

**Figure 2 brainsci-12-01257-f002:**
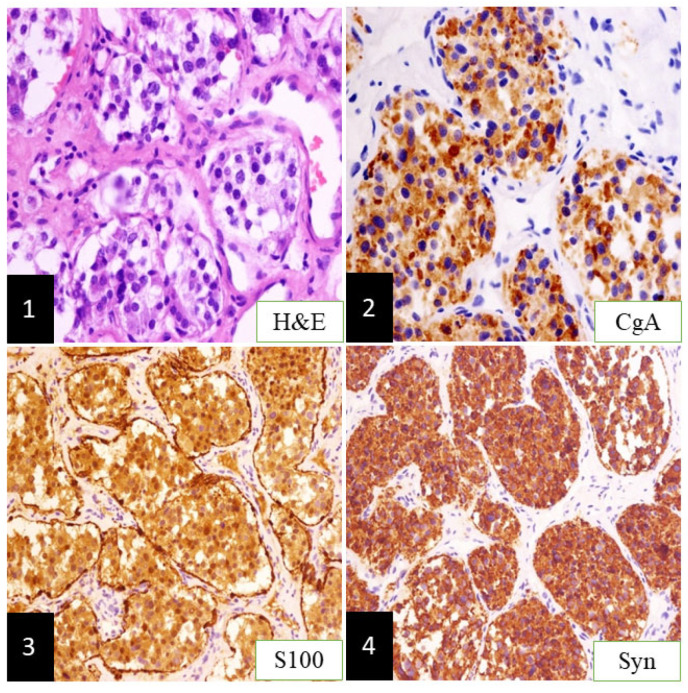
Pathological findings of intracranial lesion. (**1**): showing multiple chromaffin cytes surrounded by the vascular web and sustentacular cells; (**2**), (**3**), and (**4**): displaying immunohistochemistry results where CgA, S100, and Syn expression were positive (paraganglioma, magnification ×100).

**Figure 3 brainsci-12-01257-f003:**
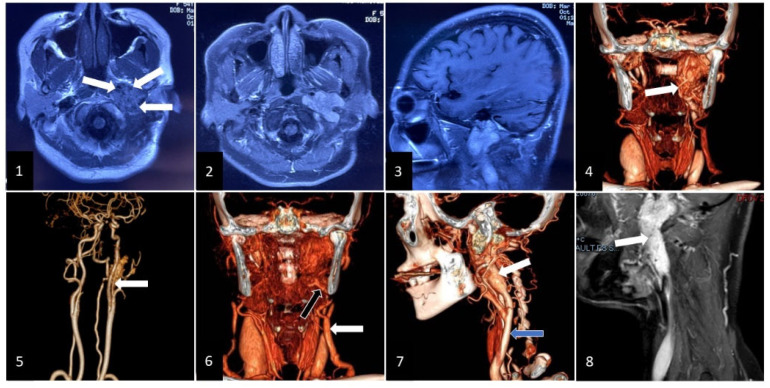
Images show radiographic results of the extracranial lesion. (**1**): showing the mass lesion with vascular flow void in left jugular foramen. White arrow: tumor with vascular flow void; (**2**) and (**3**): displaying the enlarged extracranial lesion on CE-MRI; (**4**) and (**5**): showing the feeding arteries from the external carotid artery on computed tomography angiography. White arrow: feeding arteries; (**6**): displaying the draining veins flowing into the external jugular vein. White arrow: external jugular vein. Black arrow: draining vein; (**7**): extracranial lesion being closely associated with internal jugular vein. White arrow: extracranial tumor. Blue arrow: internal jugular vein; (**8**): displaying the internal jugular vein with occlusion. White arrow: extracranial lesion and occluded internal jugular vein.

**Figure 4 brainsci-12-01257-f004:**
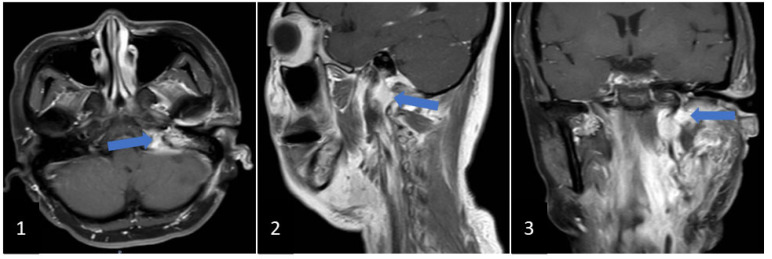
Images show the postoperative CE-MRI from the second surgery. (**1**), (**2**), and (**3**): presenting the reduced residual lesion behind the internal carotid artery. Blue arrow: residual lesion.

## Data Availability

Not applicable.

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
