# Peer review of "Staged Surgery for Intra-Extracranial Communicating Jugular Foramen Paraganglioma: A Case Report and Systematic Review"

_brainsci, 2022, doi:10.3390/brainsci12091257_

Round 1
Reviewer 1 Report
This is a case report of an intra-extra cranial communicating jugular foramen paraganglioma which was resected surgically and reoperated sequentially. The authors suggested modified surgical strategy. This is an interesting suggestion and can be considered as a potential publication if the authors address the suggestion. The suggestion is below.
Abstract:
1. Please spell out “LCN” when the authors use this for the first time.
2. Diagnostic processes are missing. I suppose that the diagnosis was made by Imaging given the lesion location and risk of complications such as massive bleeding. Please address this briefly.
3. There are some grammatically incorrect parts. Assistance from a colleague experienced in medical terminology and translation or a technical manuscript editor/service would be helpful.
Introduction:
1. Short and well-summarized introduction
2. There are some grammatically incorrect parts.
Clinical case:
1. Line 47, there is no imaging evidence of brainstem compression (Figure1). If the authors have more appropriate imaging which shows brainstem compression, it would be nice. If there is no evidence, please reword the sentence “brain stem and cerebellum were 47 slightly compressed (figure 1 and 2).”
2. Diagnostic process is missing as mentioned in the abstract. Please mention this.
3. Imaging explanation is relatively lacking. How does the lesion look on conventional MRI sequences such as T1WI and T2WI which can depict flow-void or necrotic/cystic changes?
4. Figure 3 does not show flow-void, which was pointed out by a white arrow. Please check the images with the radiologists in your institution.
5. Line 52-54, was Succinate dehydrogenase mutation checked? Hereditary head and neck paraganglioma was linked to Succinate dehydrogenase mutation. How about family history? If family history is negative, the mutation is less likely.
6. Is there any reason why radiotherapy was not performed after the first operation?
7.
Figures 1 and 3 are not photographs but MRI. Please reword them. Figure 3 (H) does not show occlusion because this is 3D reconstructed image. Please show 2D CTA or CTV to show venous occlusion. Figure 4A is fat-sat CE-MRI. Please consistent images. Otherwise, the authors use CE-MRI in Fig. 4 B and C.
8. There are some grammatically incorrect parts.
Discussion
1. Please make an imaging section to mention how paraganglioma is visualized in MRI and CT. This part is essential for the diagnosis. “Woolen S et al. Paragangliomas of the Head and Neck. Neuroimaging Clin N Am. 2016;26(2):259‐278. doi:10.1016/j.nic.2015.12.005” would be helpful.
2. Recently, perfusion MRI and diffusion-weighted images are used for differential diagnosis, treatment assessment, and gene mutation. Regarding, differentiation from other tumors, 1. “Diagnostic Role of Diffusion-Weighted and Dynamic Contrast-Enhanced Perfusion MR Imaging in Paragangliomas and Schwannomas in the Head and Neck. AJNR Am J Neuroradiol. 2021;42(10):1839-1846. doi: 10.3174/ajnr.A7266” and 2. “MR diffusion and dynamic-contrast enhanced imaging to distinguish meningioma, paraganglioma, and schwannoma in the cerebellopontine angle and jugular foramen. J Neuroimaging. 2022;32(3):502-510. doi: 10.1111/jon.12959” would be helpful.
Regarding assessment of radiotherapy effect, Diffusion-weighted and dynamic contrast-enhanced MRI to assess radiation therapy response for head and neck paragangliomas. J Neuroimaging. 2021;31(5):1035-1043. doi: 10.1111/jon.12875.
Regarding succinate dehydrogenase mutation, Assessment of MR Imaging and CT in Differentiating Hereditary and Nonhereditary Paragangliomas. AJNR Am J Neuroradiol. 2021;42(7):1320-1326. doi: 10.3174/ajnr.A7166. would be helpful. Please summarize them briefly in the imaging section, with the 4 suggested papers.
3. Please mention the reason why radiotherapy was not performed in the first surgery? The main purpose of radiation therapy is to control the size of the paragangliomas. This should be considered if there is apparently a residual tumor in the neck portion.
4. There are some grammatically incorrect parts.
Conclusion:
1. Line 179-180, “Refer to preoperative LCNs deficit and tympanic cavity involvement in IECJFP patients, we could determine specific and personalized staged surgical strategy to improve 180 long-term QoL.” This sentence is grammatically incorrect. Please correct it.
Author Response
Please see the attachment. thank you for your attention.
Reviewer 2 Report
Dear authors,
Congratulations on the case management and case presentation. I agreed to paper publishing in its present form.
Author Response
Dear reviewer,
Thank you very much for your kind work and consideration on publication of our paper. We would like to express our great appreciation to you.
Thank you and best regards.
yours sincerely,
Yanbing Yu
Reviewer 3 Report
This is a report of a 48-year-old female with Communicating Jugular Foramen Paraganglioma by Li et al. The authors have presented an interesting case and strived to offer a multi-stage approach to surgical management of patients with Communicating Jugular Foramen Paraganglioma. However, a thorough evaluation of the manuscript and its message cannot be performed. This paper has many inappropriate uses of words and syntax errors, and is generally hard to follow. As a result, one should guess the true meaning the authors intended on many occasions throughout the manuscript.
Nevertheless, it is evident that the authors have put so much effort into composing this article and their case (and the methods they are advocating for) have merit for further consideration for publication. Thus, I would suggest that this manuscript should be rewritten, preferably by a native editor, and then evaluated for publication.
Author Response
Dear reviewer,
Thank you very much for your kind work and consideration on publication of our paper. We haved corrected more errors of words, syntax and grammar in compliance with your suggestions. We would like to express our great appreciation to you.
Thank you and best regards.
yours sincerely,
Yanbing Yu
